# Dynamic Analysis of Physicochemical Properties and Polysaccharide Composition during the Pile-Fermentation of Post-Fermented Tea

**DOI:** 10.3390/foods11213376

**Published:** 2022-10-26

**Authors:** Yan Luo, Zhenjun Zhao, Hujiang Chen, Xueli Pan, Risheng Li, Dewen Wu, Xianchun Hu, Lingling Zhang, Huawei Wu, Xinghui Li

**Affiliations:** 1College of Horticulture and Gardening, Yangtze University, Jingzhou 434025, China; 2Hubei Dongzhuang Tea Co., Ltd., Xianning 437300, China; 3College of Life Science, Yangtze University, Jingzhou 434025, China; 4College of Horticulture, Nanjing Agricultural University, Nanjing 210095, China

**Keywords:** polysaccharide, post-fermented tea, pile-fermentation, UHPLC-Q-TOF-MS/MS, correlation

## Abstract

Ultra-high performance liquid chromatography-quadrupole-time of flight tandem mass spectrometry (UHPLC-Q-TOF-MS/MS) was used to study the diversity of tea polysaccharides and the dynamic changes in the physicochemical indexes of tea samples. FT-IR spectra and the free radical scavenging ability of tea polysaccharides, during pile-fermentation of post-fermented tea, were analyzed. The results showed that 23 saccharide co mponents in tea polysaccharides were identified: these belonged to 11 monosaccharides, 5 oligosaccharides, and 6 derivatives of monosaccharides and oligosaccharides. The abundance of oligosaccharides decreased gradually, while monosaccharides, and derivatives of monosaccharides and oligosaccharides increased gradually with the development of pile-fermentation. According to the differences in polysaccharide composition and their abundance, the tea polysaccharide samples extracted from different pile-fermentation stages could be clearly classed into three groups, W-0, W-1~W-4 and W-5~C-1. The pile-fermentation process affected the yield, the content of each component, FT-IR spectra, and the DPPH free radical scavenging ability of tea polysaccharides. Correlation analysis showed that microorganisms were directly related to the changes in composition and the abundance of polysaccharides extracted from different pile-fermentation stages. The study will further help to reveal the function of tea polysaccharides and promote their practical application as a functional food.

## 1. Introduction

Post-fermented tea (dark tea) has a long history of over 500 years [1]. It is considered one of the most popular tea products among the six kinds of Chinese tea around the world, and only exists in China [2]. Dark tea is made from the fresh leaves or mature shoots of tea plants (*Camellia sinensis* var.) through the main processing technologies of fixation of enzymes, rolling, pile-fermentation, steaming and pressing, and drying [3,4,5]. Due to their unique flavor quality, Pu-erh tea, Fuzhuan brick tea, Qingzhuan brick tea, and Liubao tea are recognized as the four most famous post-fermented teas, and are widely consumed by tea consumers from Macao, Hong Kong, Taiwan, Southeast Asia, mainland China, and other countries and regions [6].

Pile-fermentation is the key process for developing the unique qualities of post-fermented tea, such as the black-brown, oily color of the dried tea; the yellow-green to orange-red color of the tea soup; the special “stale” aroma; and the mellow, sweet taste [7]. Differences exist in the pile-fermentation process of post-fermented tea in different producing areas. For example, the pile-fermentation process for Hubei Qingzhuan brick tea is often divided into two stages: artificial rapid post-fermentation and natural storage; the rapid post-fermentation process lasts only 2–3 months to provide suitable temperature and humidity for microorganisms to participate in the quality formation of post-fermented tea. In contrast, the natural storage process, also known as the natural aging process, lasts for 1–2 years, and aims to promote the natural conversion of quality components in post-fermented tea under relatively low humidity. Despite differences in the process, the essence of pile-fermentation of post-fermented tea is the same in general. During the process of pile-fermentation, multiple bioactive substances in the tea substrate, such as tea polyphenols, amino acids, saccharides, and purine base etc., undergo a series of complicated biochemical reactions. These include oxidation, polymerization, decomposition, hydrolysis, methylation, and glycosylation under the enzymatic actions of extracellular enzymes produced by dominant microorganisms and the humid and hot conditions [7]. For example, catechins, especially ester catechins, are oxidized or polymerized with other organic molecules to form theaflavins, thearubigins, and theabrownins under the enzymatic action of polyphenol oxidase, laccase, or peroxidase [8]. These ingredients directly affect the formation of taste and the color quality of post-fermented tea [2]. Linolenic acid was also reported to be transformed into major flavor-active compounds, such as β-carotene, (E)-β-Ionone, etc., under the enzymatic action of lipoxygenase, hydroperoxide lyase, etc. [9], eventually forming the unique quality characteristics and healthcare functions of post-fermented tea.

It was demonstrated that the drinking of post-fermented tea leads to various biological effects and activities, including being antioxidant [10], and having anti-obesity effects [11] and antidiabetic properties [12]. It even exerts preventive and therapeutic roles for many diseases, such as hyperlipidemia, hypertension, and cancer [1]. Recently, tea polysaccharides (TPS) are widely regarded as a main natural bioactive substance in post-fermented tea, and was reported to have many kinds of healthcare functions, of which the antidiabetic effect is the most relevant [13]. Studies showed that tea polysaccharides are water-soluble acidic polysaccharides with a complex composition and structure, and have diversified physiological activity. They are mainly composed of monosaccharides, amino acids, proteins, and inorganic elements [14]**.** It is generally believed that the content, composition, and structure of tea polysaccharides change dynamically during tea processing, because it was demonstrated that the molecular weight and monosaccharide composition of tea polysaccharides varied with different tea categories, drying methods, and processing methods [15]. For example, the structural properties and biological activities of twelve crude polysaccharides extracted from six categories (dark tea, black tea, oolong tea, white tea, yellow tea, and green tea) were evaluated by Guo et al. [16]. Their results revealed that extraction yields, chemical composition, molecular weight, and compositional monosaccharides of tea polysaccharides varied among the six categories of tea; in particular, Pu-erh tea polysaccharides (extracted from the most representative post-fermented tea) had the highest extraction yield, contained the highest total phenolic and protein contents, and also exhibited the best antioxidant activities. This demonstrated that the pile-fermentation process may play an important role in affecting the content, composition, structure and function of tea polysaccharides in tea. Changes in the main physicochemical indexes during the pile-fermentation of post-fermented tea are often reported; however, the dynamic changes in content, saccharide components, structural features, and biological activity of tea polysaccharides during the pile-fermentation of post-fermented tea, and the correlation between physicochemical indexes of different pile-fermentation stages and saccharide components of tea polysaccharides, are rarely reported.

Dynamic analysis systematically analyzes the changes in the physical and chemical properties, polysaccharide composition, polysaccharide content, antioxidant activity, and other elements and functions, during the pile-fermentation of Qingzhaun brick tea; studying the changes at each stage of the piling reveals the relationship between these changes in the piling process.

In this study, the tea samples were collected in batches from different stages of pile-fermentation of Hubei Qingzhuan brick tea (a representative tea product of post-fermented tea). The samples were then studied for the purpose of a systematic evaluation of the physicochemical indexes of the different pile-fermentation stages of Hubei Qingzhuan brick tea. The changes in polysaccharide composition, structure, and antioxidant activity of different pile-fermentation stages of Hubei Qingzhuan brick tea were also systematically evaluated. The results can be helpful in revealing the influence of the pile-fermentation process of post-fermented tea on the content, composition, structure, and function of tea polysaccharides, to explore the correlation between the changes in physicochemical indexes of tea samples, and the formation of tea polysaccharides. The results will also provide theoretical reference for more efficient application of tea polysaccharides in post-fermented tea in the development of functional foods.

## 2. Materials and Methods

### 2.1. Materials and Reagents

Tea samples were collected from different pile-fermentation stages of Qingzhuan brick tea production at the Hubei Dongzhuang Tea Co., LTD. (Xianning, China) Crude sun-dried tea with one bud and five or six leaves was used as the raw material for the pile-fermentation of Qingzhuan brick tea, and labeled “W-0”. Tea leaves were naturally pile-fermented after being sprayed with water; the natural aging process was carried out after 7 weeks of pile-fermentation. During a pile-fermentation of about 49 days, samples were gathered once a week and labeled W-1 to W-7. Further samples were taken after 1 month of the natural aging process and labeled C-1. Tea samples were taken at about 10 cm from the top of the tea pile, and each sample was repeated 3 times. After sampling, the samples were mixed evenly under sterile conditions and stored at −80 °C for further study.

Unless stated, all chemical reagents were analytical grade and water was purified on a Milli-Q system (Millipore, Bedford, MA, USA). The solvents used for LC-MS analysis were of mass grade. MS-grade solvents, including acetonitrile and methanol, etc., were obtained from Tianjin Siyou (Tianjin City, China). Pure standard reagents were purchased from Sigma (St. Louis, MO, USA).

### 2.2. Samples Physicochemical Parameters Analysis

For pH Measurements, 5 g of tea powder was added into 10 mL of distilled water and mixed by hand at 5 min intervals over 30 min. The pH of the suspension was measured using a pH meter (provided from shanghai AURORA, Co., Ltd., Shanghai, China) [17]. The color difference in tea samples was measured by a colorimeter (provided from Shenzhen 3 nh Co., Ltd, Shenzhen, China) according to the method described by Wu et al. [12,18]. Protein content was determined according to the method of Coomassie brilliant blue colorimetry, as described by Song et al. [19]. The contents of tea polyphenols, total free amino acids, caffeine, catechin, and water extract were determined according to the methods described in the corresponding national standards issued in 2013 [20]. The contents of flavonoids and soluble sugars were determined using the aluminum trichloride colorimetric method [21] and the anthrone-sulfuric acid colorimetric method [22], respectively. The content of tea polysaccharides and theabrownins was ascertained as described by Zhuang et al. [23] and Roberts et al. [24].

### 2.3. Microbiological Analysis

The microbial population of pile-fermented tea samples was enumerated using the method of plate dilution, as described previously by Zhao et al. [25]. The diluted suspension of tea samples was spread on the surface of the medium (Dichloran Rose-Bengal Chloramphenicol Agar medium for fungi and Beef extract peptone medium for bacteria). All plates were incubated at 25~30 °C for up to 7 days, with bacteria being enumerated after 3 days and fungi after 5~7 days.

### 2.4. Preparation of Polysaccharides and Analysis of Their Main Components

The polysaccharides of tea samples in the pile-fermentation process of post-fermented tea were extracted by the method described previously ([16,26] with some modifications. Briefly, 50 g of dry meshed tea powders were precisely weighed and treated with 80% ethanol for 24 h to remove most small molecular substances. After the supernatant was removed, the residues were dried in air and then extracted with distilled water at 70 °C for 2 h (3 times) to acquire the polysaccharides. The water extract of polysaccharides from tea samples was concentrated by rotary evaporation, freeze-dried, and stored at −5 °C for subsequent studies.

The extraction yields of polysaccharides were determined by the weighing method [16]. Neutral sugar content was measured using the anthrone-sulfuric acid method, using D-glucose as standard [22]. Total protein was analyzed with the Bradford method, using bovine serum albumin as the standard [27]. Total polyphenol and amino acid content in the polysaccharide extractions was measured using the Folin–Ciocalteu method and the ninhydrin method [16].

### 2.5. Analysis of Polysaccharides by Fourier-Transform Infrared Spectrometry

Fourier-transform infrared spectrometry of the polysaccharide extraction was conducted using a Fourier-transform infrared spectrophotometer (FTIR-680, Tianjin RuiAn, China). The freeze-dried powder of the polysaccharide extraction was evenly mixed with KBr powder and then pressed into pellets for FTIR measurement at a frequency range of 400—4000 cm^−1^. For the specific method, we referred to Wang et al. [28].

The estimation of the degree of esterification (DE) was made on the basis of the equation below [16].
DE%=A1735A1735+A1630×100

### 2.6. Analysis of Polysaccharide Composition by Ultra-High Performance Liquid Chromatography-Quadrupole-Time of Flight Tandem Mass Spectrometry (UHPLC-Q-TOF-MS/MS)

An appropriate amount of freeze-dried polysaccharide powder was dissolved in a pre-cooled methanol/acetonitrile/aqueous solution (2:2:1, *v*/*v*). The solution was mixed by vertexing, left to stand at −20 °C for 10 min, and the supernatant was dried under vacuum. Four mg of dried product was redissolved in 1.5 mL hydrochloric acid aqueous solution (1 mol/L). The mixture was heated in a 100 °C water bath for 1 h and then cooled to room temperature. Residual hydrochloric acid was removed by drying and the dried product was redissolved in deionized water. The solution was filtered through 0.45 µm filter membrane for MS spectrometry analysis [29,30].

The Agilent 1290 Infinity LC ultra-high performance liquid chromatography (UHPLC) system was used to identify polysaccharide composition. The chemical composition was separated through the C-18 (Water, Millerstown, PA, USA) chromatographic column. The flow rate was 0.4 mL/min, the column temperature was 40 °C, and the injection volume was 2 µL. Mobile phase A was Water +25 mM ammonium acetate +0.5% formic acid, and mobile phase B was methanol. The gradient elution of mobile phase B was 0~0.5 min, 5%; 0.5~10 min, 5~100%; 10.0~12.0 min, 100%; 12.0~12.1 min, 100~5%; and 12.1~16 min, 5%.

An AB Triple TOF 6600 mass spectrometer and an electrospray ion source (ESI) were used to collect the primary and secondary spectrograms of the samples. The data were collected in the positive and negative ion modes. The condition of the electrospray ion source (ESI) was ion source gas 1 (Gas1): 60; ion source gas 2 (Gas2): 60; curtain gas (CUR): 30; source temperature: 600 °C; and ion sapary voltage floating (ISVF) ± 5500 V. The TOF MS scan *m*/*z* range was 60~1000 Da; the production scan *m*/*z* range was 25~1000 Da; the TOF MS scan accumulation time was 0.20 s/spectra, and the production scan accumulation time was 0.05 s/spectra. Secondary mass spectra were obtained using information-dependent acquisition (IDA) and a high-sensitivity model.

The software MSDAIL was used for peak extraction, peak filtration, peak alignment, and peak identification, and the data matrix, including the mass-to-charge ratio (*m*/*z*), retention time, and peak area (intensity) was obtained. The identification of polysaccharide composition was performed on the data extracted using MSDAIL and other mass spectral information from the relevant literature related to tea components.

### 2.7. DPPH Free Radical Scavenging Activity Analysis of Polysaccharide from Different Stages of Pile-Fermentation

The DPPH free radical scavenging activities of the polysaccharide from different pile-fermentation stages of post-fermented tea were assayed by the method as described by [31], with a slight modification. A quantity of 1 ml of each of the tested samples at various concentrations (0.2~1.0 mg/mL) was added to 3 mL of the ethanolic DPPH solution (0.1 mmol/L), respectively. The absorbance of the mixed solution was measured at 517 nm after incubation for 30 min at room temperature in the dark. Distilled water of an equal volume was used as the positive control. The DPPH radical scavenging activity was calculated as follows:$$\mathrm{DPPH}\;\mathrm{scavenging}\;\mathrm{activity}\;\left(\%\right)=\left(1-A_{sample}/A_{control}\right)\times 100$$
where *A* sample and *A* control were defined as the absorbance of the sample and the control, respectively.

### 2.8. Statistical Analysis

All the experiments were carried out in triplicate; data are presented as the mean value plus or minus the standard deviation (SD). Differences between groups were examined with an independent sample test. The heat map and principal component analysis (PCA) was used to evaluate the relationship among the parameters of tea polysaccharides. Correlation analysis was used to reveal the relationship between physicochemical indexes and tea polysaccharide components in the process of post-fermentation tea piling. Origin 2021 statistical software was used to prepare the data graphs.

## 3. Results and Discussion

### 3.1. Analysis of Physicochemical Properties of Post-Fermented Tea during the Pile-Fermentation Dynamic Changes in Main Characteristic Index during the Pile-Fermentation Process

In this paper, tea samples from different pile-fermentation stages of Qingzhuan brick tea were studied to analyze their physicochemical characteristics. The dynamic changes in the physicochemical characteristics during the pile-fermentation process of the tea samples are shown in Figure 1.

The water content of the tea raw material (W-0) was 10.41%; the water content reached 43.00% after spraying (W-1) and decreased to about 15.00% at the late stage of the pile-fermentation of the tea samples (shown in Figure 1a). The water content decreased from 43.00% to 16.46% in the process of pile-fermentation, which was similar to the water content decreasing from 37.11% to 14.52% in the pile-fermentation process of Pu-erh tea [32]. The water content decreased rapidly in the early stage of fermentation, but slowly during the later stage. The pH of the tea samples decreased with the development of pile-fermentation: it decreased to its lowest value of 4.01 in the pile-fermentation stage for W-2, and then floated around 4.50 until the end of the first stage of pile-fermentation (W-7). At the natural aging stage of pile-fermentation, the pH of the tea samples increased significantly to 5.88 (*p* < 0.05) after one month of aging (C-1) (Figure 1b). It is generally believed that the fluctuations in tea pH are related to the growth and reproduction of microorganisms in the process of piling: acid-producing and acidophilic microorganisms are thought to closely affect the acidity of tea [33].

The color difference indexes *L**, *a**, and *b** can be used to identify the subtle differences in the lightness, and the red and yellow tones of the tea samples (the *L** value represents lightness, the higher, the brighter; the higher the *a** value, the deeper the red color; and the higher the *b** value, the deeper the yellow color). The indexes, thus, provide an important basis for evaluating the formation of tea quality during the pile-fermentation process of post-fermentation [34,35]. The color difference values (*L**, *a**, *b**) of tea samples fluctuated during the pile-fermentation process (*L** fluctuated in a downward trend; *a** and *b** fluctuated in an upward trend). This is similar to the conclusion drawn by Zhang [36] in the process of Sichuan dark tea piling: “With the progress of piling, *L** value gradually decreases, and the brightness of tea soup decreases.” The value of *a** gradually increased from negative to positive, and the color of the soup changed from “green” to “red”. The fluctuation in the *b** value increased, and the degree of yellow in the soup deepened. This indicates that the changing trend in the tea-sample color was from yellow-green, orange-yellow, and orange-red to brown-red during the pile-fermentation process (Figure 1c). The changes in the *L**, *a**, and *b** values of the tea samples were significantly correlated (*p* < 0.05) with the changes in the physical and chemical components of the tea samples [37,38]. With the content decrease in tea polyphenols, amino acids, and catechins, and the increase in theabrownin content, the *L** value of the tea samples gradually decreased, while significant negative correlations were found between the *a** and *b** values and the content of the main tea components (such as tea polyphenols, amino acids, and catechins).

The aqueous extract content of the tea samples in the pile-fermentation process decreased continuously, from 37.36% to 28.76%, with a maximum decrease of 23.55% (*p* < 0.01); this may be closely related to a series of biochemical reactions [38]. For example, as the pile-fermentation progresses, microorganisms continue to multiply in large numbers and consume the decomposed soluble compounds as carbon and nitrogen sources. Alternatively, because of degradation and oxidation activity under the high temperature and humidity conditions during the pile-fermentation process of post-fermented tea, the water extract content is reduced. There were regular dynamic changes in the pH, water content, color, and aqueous extract of the tea soup during the pile-fermentation process of the tea samples; these changes could be used as important physical and chemical characteristic indexes to measure the pile-fermentation process of post-fermented tea [36,39].

### 3.2. Dynamic Changes in Main Quality Components during the Pile-Fermentation Process

Tea polyphenols and their oxidation products, amino acids, and soluble sugars are important quality components of post-fermented tea. The transformation and accumulation of quality components are closely related to microbial activities during the pile-fermentation process of post-fermented tea [40]. A heatmap analysis of the quality components during the pile-fermentation of the tea samples is shown in Figure 2 (specific data in Appendix A). Similar to references [2,11,37], the content of tea polyphenols, catechins, flavones, amino acids, and proteins decreased continuously, while the content of theabrownins, soluble sugar, and tea polysaccharides increased, and that of caffeine was basically unchanged during the pile-fermentation process of Qingzhuan brick tea. Tea polyphenols and catechins reached the lowest value at the W-5 stage, with the maximum reduction (*p* < 0.05) in tea polyphenols and catechins reaching 59.00% and 58.29%, respectively. This may be because the phenolic compounds were continuously oxidized and degraded by extracellular enzymes secreted by microorganisms under the condition of high temperature and high humidity during the pile-fermentation process of tea samples [41]. Amino acid is an important flavor substance, manifested as the freshness of tea soup, and is of great significance in the formation and transformation of tea flavor [42]. With the progress of fermentation, the content of amino acid decreases continuously. Tea polysaccharide is a type of acid polysaccharide or acid glycoprotein bound with protein [14]. The content of tea polysaccharides reached highest value at the W-5 stage, and then began to show a downward trend. Protein and tea polysaccharides showed the opposite trend, which may be because proteins were involved in biochemical reactions during the pile-fermentation process; for example, hydrolysis to produce amino acids, or binding to glycogen to produce tea polysaccharides. Theabrownin content gradually increased, and the highest content of theabrownins was found at C-1 with a maximum increase (*p* < 0.05) of 127.42%; this was related to the oxidation and polymerization of polyphenols during the pile-fermentation process. The content of caffeine was less affected by moisture–heat action and enzymatic oxidation during the pile-fermentation process, which may be due to its stable ring structure [38,43].

### 3.3. Analysis of Number of Microorganisms during the Pile-Fermentation of Post-Fermented Tea

The changes in microbial population during the pile-fermentation process are shown in Figure 3. The number of fungal populations fluctuated and increased in the process of pile-fermentation. The lowest value for the number of fungal populations was 0.42 × 10^7^ cfu/g at the W-4 stage, and the maximum value was 7.57 × 10^7^ cfu/g (*p* < 0.05) at the W-7 stage; the number of fungal populations then decreased significantly at the C-1 stage. The number of bacteria populations increased slowly from the W-0 stage to the W-2 stage, then increased rapidly after the W-2 stage, reaching a maximum value of 20.30 × 10^8^ cfu/g (*p* < 0.05) at the W-4 stage. At the W-5 stage, the number of bacteria populations decreased rapidly to 0.69 × 10^8^ cfu/g, and was then maintained at a low level during the remaining stages of pile-fermentation. During the whole process of pile-fermentation, the number of fungal populations increased in the middle and late stages of pile-fermentation. The number of bacterial and fungal populations fluctuated with regularity, and their importance in the formation of post-fermented tea quality in the process of pile-fermentation was confirmed by many studies. For example, thermophilic strains can produce cellulase, amylase, and tannase; increase the content of soluble sugar; and reduce the content of ester catechin, thereby reducing the bitter taste of tea [44]. Furthermore, yeast microorganisms can secrete enzymes that degrade phenol and produce sweet xylitol substances [3].

### 3.4. Chemical Components of Polysaccharides in Post-Fermented Tea during Pile-Fermentation

The extraction yield, tea polyphenols, amino acid, neutral sugar, total protein, and degree of esterification (DE) of polysaccharides extracted from the tea samples in the process of pile-fermentation are summarized in Table 1. The extraction yield of polysaccharides in the tea samples from different pile-fermentation stages increased gradually and reached a maximum of 11.55% (*p* < 0.05) in the W-6 stage of pile-fermentation. Subsequently, the extraction yield began to decline, and the extraction yield of polysaccharides was consistent with the changes in microbial population during the pile-fermentation process. Consistent with Zhu’s research [45], the thermophilic strain can produce cellulase, which degrades cellulose to produce glucose; it can also produce amylase, which hydrolyzes starch into oligosaccharides. speculation that the formation of polysaccharides in tea may be affected by microbial metabolism and the secretion of extracellular enzymes during the pile-fermentation process of tea samples. Neutral sugar and protein were the core components of tea polysaccharides [26]. Changes in the neutral sugar content in the polysaccharide extract showed the same trend as the changes in polysaccharide extract yields in the pile-fermentation process, and the maximum value was reached at W-6 (41.41%) stage of pile-fermentation. However, the content of protein in the polysaccharide extract was low and less affected by the tea pile-fermentation process. These polysaccharide extracts also contained a certain amount of tea polyphenols, amino acids, and proteins (Table 1), consistent with the result of the polysaccharide composition obtained from the study of Pu-erh tea polysaccharides by Mao et al. [26]. This indicates that polysaccharide molecules are easy to form more stable macromolecules with tea polyphenols and amino acids, through hydrogen bonds or other links. In addition, the degree of esterification of polysaccharide extracts was evaluated using infrared spectroscopy. Interestingly, the changes in the esterification degree of the polysaccharide extracts were also affected by the pile-fermentation process, and the changing trend is consistent with that of the polysaccharide extraction yields and neutral sugar content.

### 3.5. Infrared Spectroscopic Analysis of Polysaccharides in Post-Fermented Tea during Pile-Fermentation

The FT-IR spectra of polysaccharides extracted from tea samples at different pile-fermentation stages of post-fermented tea are shown in Figure 4; the assignment of the main FT-IR bands of polysaccharides is listed in Table 2. In general, all polysaccharide samples exhibited very similar FT-IR spectra, except for a few weak bands. It can be seen that a broadly stretched band at around 3414~3442 cm^−1^ represents the characteristic absorption of a hydroxyl group. The presence of hydrogen bonds [44], two weak bands at around 2929~2949 cm^−1^ and 768–822 cm^−1^, suggest the presence of C-H stretching and C-H bending, respectively. The relatively strong peaks at around 1609–1638 cm^−1^ and 1419–1436 cm^−1^ are due to the C=O stretching of the carboxylate groups (indicating the presence of proteins) [26] and the C-O stretching vibration peak [44], respectively. The bands at around 1080–1107 cm^−1^ and 1026–1053 cm^−1^ for all polysaccharides are the characteristic spectra of pyranoside and sugar rings [26]. These results suggest that all the tea polysaccharides are composed of saccharides and proteins, which is consistent with Wei et al. [45], who proposed that the essence of tea polysaccharide is a glycoprotein, which is tightlyis closely bound to the sugar chain by N- or O-covalent bonds. This is also consistent with studies by Kang [46] and Chen [47], who proposed similar studies of β-pyranoside linkage and protein-bound polysaccharides.However, at the same time, the polysaccharides extracted from the tea samples at different pile-fermentation stages of post-fermented tea have their own characteristic bands; for example, the weak bands at around 2859–2882 cm^−1^, 1327 cm^−1^, and 1239–1250 cm^−1^ were only found in the polysaccharides extracted from W-0/W-1/W-3/W-5, W-0/W-2/W-4~W-7, and W-0~W-5/W-7/C-1, respectively. The differences in the FT-IR spectra of the polysaccharides suggest that the FT-IR spectra of the polysaccharides extracted from the tea samples at different pile-fermentation stages of post-fermented tea were also affected by the pile-fermentation process.

### 3.6. Mass Spectrometry Analysis of Tea Polysaccharide Components during the Pile-Fermentation of Post-Fermented Tea

Saccharide components in tea polysaccharide extracts from tea samples at different pile-fermentation stages were analyzed by UHPLC-Q-TOF-MS/MS [30]. The results show that twenty-three carbohydrate components were identified as belonging to: eleven monosaccharides, namely, d-arabinose, l-rhamanose, d-glucose, d-mannose, mannitol, d-glucosamine, galactaric acid, 2-dehydro-d-gluconic acid, gluconic acid, D-arabitol and 1,5-anhydro-d-glucitol; five oligosaccharides, namely, sucrose, beta-maltose, raffinose, maltotriose, and stachyose; and six derivatives of monosaccharides and oligosaccharides, namely, glucose 6-phosphate, α-d-galactose 1-phosphate, d-fructofuranose 6-phosphate, 1,6-digalloyl-beta-d-glucopyranose, 1,3,6-tri-*o*-galloyl-beta-d-glucose and 1,2,3-tri-*O*-galloyl-beta-d-glucose.

In general, according to the dynamic change in the relative polysaccharide component abundance values, it can be judged that oligosaccharide abundance (17.43%)>monosaccharide abundance (6.64%)>monosaccharide and oligosaccharide derivatives (0.73%) at the initial stage (W-0) of tea pile-fermentation. Hereafter, the abundance of oligosaccharides began to decrease, and monosaccharides and derivatives of monosaccharides and oligosaccharides began to increase. At the end of the tea pile-fermentation stage (C-1), the abundance values of the saccharide composition of tea polysaccharides were: monosaccharides (14.23% >derivatives of monosaccharides and oligosaccharides (3.14%)>oligosaccharides (1.87%). The dynamic changes in polysaccharide components were basically consistent with the changes in the microbial population in the process of tea pile-fermentation, which means that microbial activity had an important influence on the formation of tea polysaccharides. It is speculated that, in the process of tea pile-fermentation, microorganisms may participate in polysaccharide transformation during tea heap fermentation. Polysaccharide transformation may include derived oligosaccharides and monosaccharides by enzyme action (for example, phosphorylation) to take advantage of carbohydrates. The specific mechanism is unclear, but an analysis [44] of enzyme production by thermophilic bacteria in sun-dried green tea found that thermophilic strains during the pile-fermentation of the tea can produce cellulase, amylase, and tannin enzymes. At the same time, the involvement of thermophilic bacteria in tea pile-fermentation resulted in the increase in soluble sugar content and the decrease in the ester catechin content in tea. Cellulase can degrade cellulose to produce glucose, and amylase can hydrolyze starch to produce oligosaccharides, thus increase the content of soluble sugar in tea. Tannase can hydrolyze tannic acid substances and reduce the bitterness of tea.

According to the abundance of polysaccharide components, principal component and heat map analyses of polysaccharide components at different pile-fermentation stages were carried out, as shown in Figure 5a,b. According to Figure 5a, the contribution rates of PC1 and PC2 are 42.7% and 31.1%, respectively. The samples from W-1, W-2, W-3, and W-4; and W-5, W-6, W-7 and C-1, had obvious aggregation phenomena, indicating that the tea polysaccharide samples had a certain similarity in composition and could be obviously aggregate into three parts, W-0, W-1~W-4 and W-5~C-1. The components of tea polysaccharide samples at the W-0 stage mainly included d-glucose, d-glucosamine, 2-dehydro-d-gluconic acid, gluconic acid, sucrose, beta-maltose, raffinose, and maltotriose, which contributed greatly to PC1. The components of tea polysaccharide samples at stages of W-1~W-4 mainly included d-glucose, d-glucosamine, 2-dehydro-d-gluconic acid, gluconic acid, sucrose, raffinose, maltotriose, glucose 6-phosphate, α-d-galactose 1-phosphate, and 1,3,6-tri-*O*-galloyl-beta-d-glucose, which contributed greatly to PC2. The components of the tea polysaccharide samples at the stage W-5~C-1 mainly included d-arabinose, l-rhamanose, d-glucose, d-mannose, d-glucosamine, 2-dehydro-d-gluconic acid, d-arabitol, sucrose, glucose 6-phosphate, α-d-galactose 1-phosphate, d-fructofuranose 6-phosphate, and 1,6-digalloyl-beta-d-glucopyranose, which were negatively correlated with PC1 and PC2. In Figure 5b, the horizontal axis represents tea polysaccharide samples extracted from different pile-fermentation stages, while the vertical axis represents different polysaccharide components. Polysaccharide samples with similar compositions were grouped together. Color-coded scale grading from red to blue corresponds to a shifting of the polysaccharide compound content from low to high. In Figure 5b, the relative content of polysaccharide components in different polysaccharide samples is represented by different colors. Red represents a high similarity in the content of polysaccharide components in different polysaccharide samples, while blue represents a low similarity. Considering comprehensively the composition and content of tea polysaccharides, the tea polysaccharide samples extracted from the different pile-fermentation stages were classed into four groups: W-0, W-1~W-2, W-3~W-6, and W-7~C-1, which were closely related to the activity of microorganisms in the pile-fermentation process.

The principal components and heat map analyses of tea polysaccharide samples based on monosaccharide composition and abundance are shown in Figure 6a1,a2. The contribution rates of PC1 and PC2 were 48.4% and 30.0%, respectively. Tea polysaccharide samples extracted from different pile-fermentation stages were classed into five groups: W-0, W-1~W-2, W-3, W-4, and W-5~C-1, according to their monosaccharide composition and abundance. In general, the monosaccharide substances in the tea polysaccharides were more abundant, and the content of each monosaccharide showed an increasing trend with the development of pile-fermentation.

The principal components and heat map analyses of tea polysaccharide samples based on the composition and abundance of oligosaccharides are shown in Figure 6b1,b2. The contribution rates of PC1 and PC2 were 99.5% and 0.5%, respectively. With the progress of pile-fermentation, the tea polysaccharide samples gradually gathered together and could be divided into three groups: W-0, W-1~W-7, and C-1, which was the same as the three-stage classification of the pile-fermentation process (early pile-fermentation stage, middle and late pile-fermentation stages, and aging stage) based on the conversion of quality components, but different from that of the monosaccharides.

The principal components and heat map analyses of tea polysaccharide samples based on the composition and abundance of the derivatives (of monosaccharide and oligosaccharide) are shown in Figure 6c1,c2. The contribution rates of PC1 and PC2 were 59.6% and 26.9%, respectively. According to the composition and abundance of monosaccharide and oligosaccharide derivatives, the tea polysaccharide samples extracted from the different pile-fermentation stages were classed into four groups: W-0, W-1~W-2, W-3~W-6, and W-7~C-1, which was similar to that of the monosaccharides.

### 3.7. DPPH Free Radical Scavenging Activity of Tea Polysaccharides during Pile-Fermentation of Post-Fermented Tea

The DPPH radical determinations often used to assess the antioxidant capacity of natural products by testing their free radical scavenging ability [55]. The scavenging ability of tea polysaccharides, extracted from different pile-fermentation stages, of DPPH radicals is shown in Figure 7a,b. All the tea polysaccharide extracts at the different pile-fermentation stages exhibited effective scavenging ability of DPPH radicals but, in general, their free radical scavenging activity decreased gradually with pile-fermentation (Figure 7a). Tea polysaccharides extracted from raw material samples had the strongest free radical scavenging activity, which was 74.08% at the W-0 stage, and this activity decreased significantly after pile-fermentation, to 46.46% at the C-1 stage. The decrease in free radical scavenging activity may be related to the decrease of polyphenols content and sugars in the tea polysaccharide extracts [56]. For instance, the free radical scavenging activity of tea polysaccharide at the W-2 stage was significantly higher than that of W-1 and W-3, possibly because the tea polysaccharide extract at the W-2 stage contained more tea polyphenols, as shown in Table 1. The free radical scavenging activity of tea polysaccharides at the W-5 stage was also significantly higher than that of the adjacent stages (W-4, W-6); this may be related to the significant increase in the abundance of reducing saccharides in tea polysaccharide components at the W-5 stage, as shown in Table 3. In addition, the free radical scavenging activity of tea polysaccharides extracted from different pile-fermentation stages also showed in the manner of dose-dependency. In the concentration range of 0.2–1 mg/L (shown in Figure 7b), it was found that the free radical scavenging activity of tea polysaccharide extracts increased with the increase in concentration. This was consistent with the conclusion of Shi [57] and Zhang et al. [58], in that the reducing power of the tea polysaccharide samples was dose-dependent on its concentration.

### 3.8. Correlation Analysis between Composition, Biological Activities of Tea Polysaccharide, and Physicochemical Characteristics during the Pile-Fermentation of Post-Fermented Tea

The main physical and chemical indexes of post-fermented tea (including water content, pH, water extract content, color difference, number of microbial populations, tea polyphenols, and tea theabrownins, etc.) change regularly during the pile-fermentation process [59]. There are important physical and chemical characteristic marks for measuring the pile-fermentation process of post-fermented tea. In order to reveal the influence of the pile-fermentation process on the formation of tea polysaccharides in post-fermented tea, the correlation between the chemical components of tea polysaccharides, and the main physical and chemical indexes of tea samples in the pile-fermentation process, were analyzed in this study; the results are shown in Figure 8. The color of the grid in Figure 8 represents the correlation between two parameters: red represents a positive correlation, blue represents a negative correlation, and white represents no significant correlation; the darker the color, the stronger the correlation.

As shown in Figure 8, most of physical and chemical indexes of post-fermented tea, including tea polyphenols, catechins, flavonoids, amino acids, aqueous extract, total protein, and the color difference *L** value were significantly negatively correlated with the abundance of monosaccharides and positively correlated with oligosaccharide abundance. Furthermore, some characteristic components, such as theabrownins, were positively correlated with the abundance of several monosaccharides (d-arabinose, l-rhamanose, d-Mannose, d-Arabitol) and negatively correlated with an abundance of all oligosaccharides (*p* < 0.05). Previous studies had, however, confirmed that microorganisms could secrete a variety of extracellular enzymes to affect the changes in tea physical and chemical indexes during the pile-fermentation of post-fermented tea; for example, cellulase and amylase secreted by thermophilic bacteria were conducive to the accumulation of tea-soluble sugar [44]. In addition, polyphenol oxidase and peroxidase secreted by fungi contributed to the oxidation of polyphenols and the formation of theabrownins [60]. Therefore, this study concludes that microorganisms are directly related to the composition and abundance changes in tea polysaccharides, while other physical and chemical indexes could only be used as indirect references to evaluate the composition and abundance of tea polysaccharides at different pile-fermentation stages. Interestingly, the fungal population was positively correlated with the abundance of d-arabinose, l-rhamanose, d-glucose, mannose, mannitol, 1,5-anhydro-d-glucitol, 1,6-digalloyl-beta-d-glucopyran-ose, and 1,3,6-tri-*O*-galloyl-beta-d-glucose, but negatively correlated with the abundance of D-glucosamine, galactaric acid, 2-dehydro-d-gluconic acid, gluconic acid, d-arabitol, sucrose, beta-maltose, raffinose, maltotriose, and stachyose d-fructofuranose 6-phosphate. The bacterial population was positively correlated with the abundance of d-glucose mannose d-glucosamine, galactaric acid, gluconic acid, d-arabitol, 1,5-anhydro-d-glu- citol, glucose 6-phosphate, α-d-galactose 1-phosphate, and 1,3,6-Tri-*O*-galloyl-beta-d-glucose, but negatively correlated with the abundance of d-arabinose, l-rhamanose mannitol, 2-dehydro-d-gluconic acid, sucrose, beta-maltose, raffinose, maltotriose, stachyose, and 1,2,3-tri-*o*-galloyl-beta-d-glucose. This means that during the process of tea pile-fermentation, microorganisms participate in the hydrolysis or decomposition of oligosaccharides (such as sucrose, beta-maltose, raffinose, maltotriose, and stachyose), preferring to use monosaccharides such as glucose as their carbon source for life activities. Other monosaccharides, such as d-arabinose, however, might not be used preferentially as carbon sources, leading to their accumulation and, ultimately, to an increase in their abundance.

In addition, the content of tea polysaccharides and soluble sugar were positively correlated with monosaccharides, but negatively correlated with oligosaccharides, indicating that oligosaccharides and their derivatives may only be an intermediate conversion product in the formation process of tea polysaccharides during the pile-fermentation process of post-fermented tea, and monosaccharides may be the core components of tea polysaccharides.

## 4. Conclusions

This study investigated the dynamic changes in the physicochemical properties and the characteristics of tea polysaccharides from different pile-fermentation stages of post-fermented tea. We found that the tea polysaccharides extracted from different pile-fermentation stages belonged to eleven monosaccharides, five oligosaccharides, and six derivatives of monosaccharides and oligosaccharides; these consisted of tea polyphenols, amino acids, neutral sugars, and total proteins. The FT-IR spectra of polysaccharides extracted from different pile-fermentation stages showed certain similarities. The results of the PCA showed that the tea polysaccharide samples extracted from different pile-fermentation stages could be clearly classed into three groups. Correlations between the different parameters showed that microorganisms were directly related to the changes in composition and abundance of polysaccharides; monosaccharides may be the core components of tea polysaccharides; and oligosaccharides and their derivatives may only be the intermediate conversion products in the formation of tea polysaccharides during the pile-fermentation of post-fermented tea.

Briefly, the study explored the dynamic changes in polysaccharide properties and their influencing factors during the pile-fermentation process of post-fermented tea. It also demonstrated that the fermentation process of post-fermented tea polysaccharide components in the tea heap is constantly changing, and has a positive relationship with microorganisms. This knowledge will further help reveal the function of tea polysaccharides and promote their practical application as a functional food. Nevertheless, the specific path of the transformation of tea polysaccharide components is not known for the time being, and requires further research.

## Figures and Tables

**Figure 1 foods-11-03376-f001:**
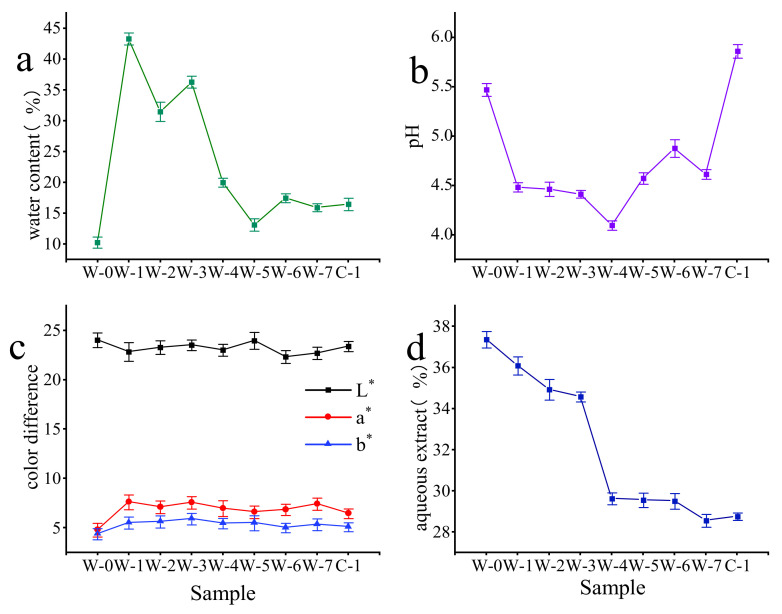
Dynamic changes in water content (**a**), pH (**b**), color difference (**c**), and aqueous extract (**d**) of tea samples during the pile-fermentation of post-fermented tea (*n* = 3).

**Figure 2 foods-11-03376-f002:**
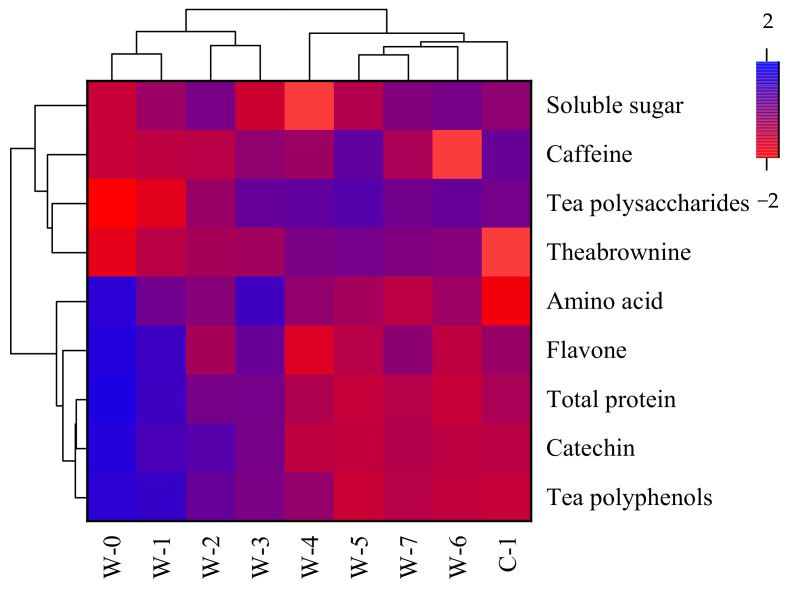
Heatmap analysis of quality components during the pile-fermentation process of post-fermented tea. Notes: Color-coded scale grading from red to blue corresponds to a shifting of the bioactive compound content from low to high.

**Figure 3 foods-11-03376-f003:**
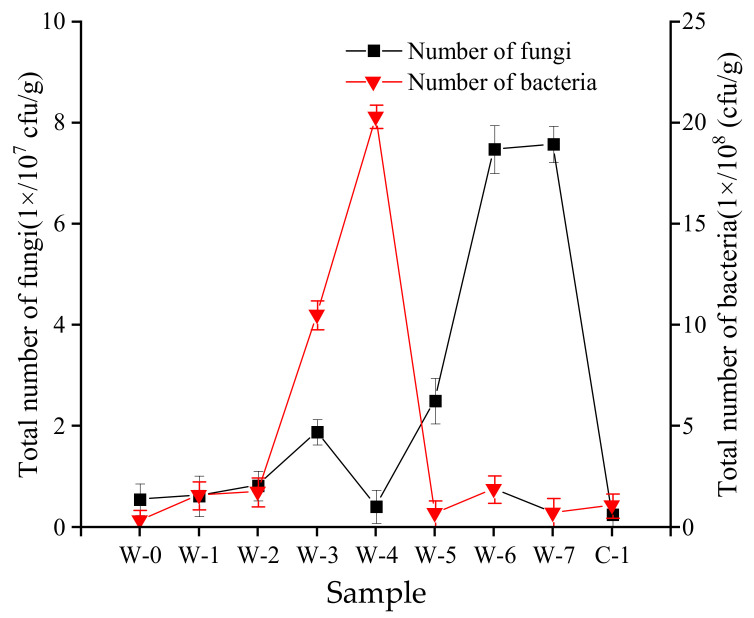
Number of microbial populations during the pile-fermentation of post-fermented tea.

**Figure 4 foods-11-03376-f004:**
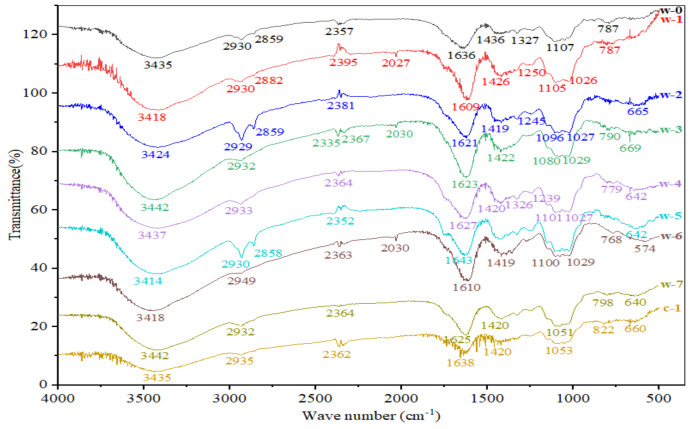
Infrared spectrum of tea polysaccharides from different pile-fermentation stages of post-fermented tea.

**Figure 5 foods-11-03376-f005:**
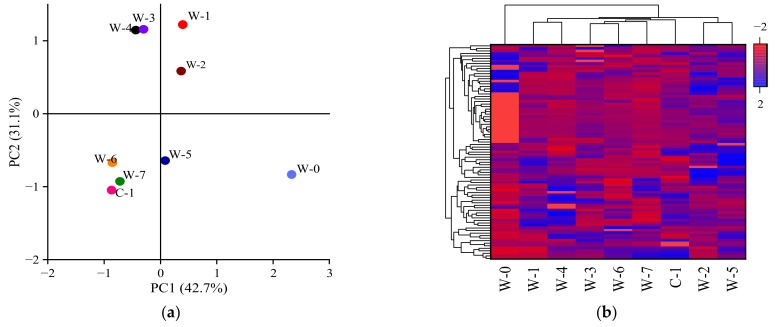
Analyses of principal components (**a**) and heat map (**b**) of polysaccharide composition in different pile-fermentation stages of post-fermented tea.

**Figure 6 foods-11-03376-f006:**
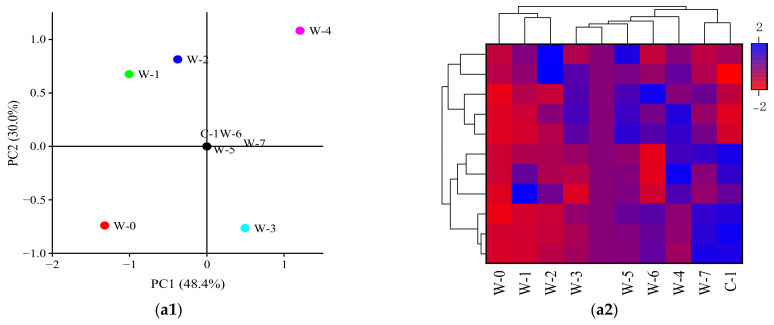
Analysis of principal components and heat map for monosaccharides (**a1**,**a2**), oligosaccharides (**b1**,**b2**) and their derivates (**c1**,**c2**) in tea polysaccharide from different pile-fermentation stages of post-fermented tea.

**Figure 7 foods-11-03376-f007:**
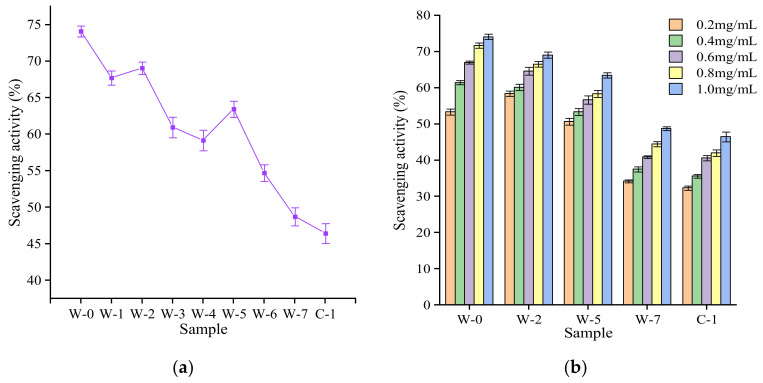
DPPH free radical scavenging activity of tea polysaccharides during the pile-fermentation of post-fermented tea. Notes: (**a**): different pile-fermentation stages (1.0 mg/mL); (**b**): different concentrations of tea polysaccharides. Each value represents the mean ± SD (*n* = 3).

**Figure 8 foods-11-03376-f008:**
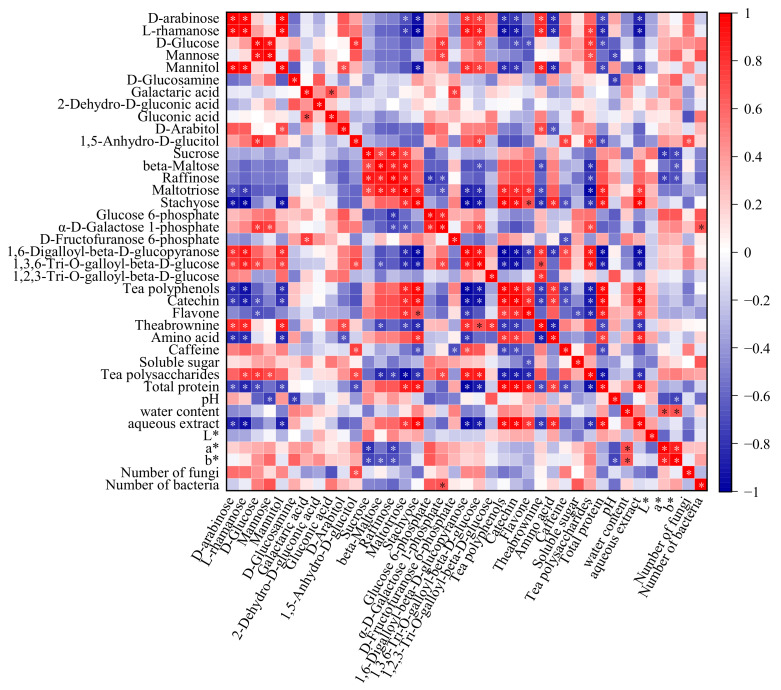
Heat map analysis of the correlation between polysaccharide composition and physicochemical properties during pile-fermentation. Significant difference: * *p* ≤ 0.05.

**Table 1 foods-11-03376-t001:** The extraction yields and chemical composition of tea polysaccharides (%).

No.	Extraction Yield *	Tea Polyphenols	Amino Acid	Neutral Sugar	Total Protein	Degree of Esterification
W-0	1.73 ± 0.38 ^f^	10.33 ± 3.04 ^a^	1.62 ± 0.12 ^c^	29.72 ± 1.67 ^h^	8.19 ± 0.11 ^b^	51.35 ± 0.12 ^g^
W-1	5.16 ± 0.34 ^e^	9.55 ± 3.48 ^c^	1.43 ± 0.26 ^b^	29.32 ± 1.08 ^h^	8.23 ± 0.16 ^b^	52.61 ± 0.05 ^f^
W-2	7.79 ± 0.82 ^d^	8.34 ± 3.20 ^b^	1.32 ± 0.14 ^c^	31.43 ± 0.93 ^e^	8.18 ± 0.00 ^b^	53.72 ± 0.03 ^e^
W-3	9.27 ± 0.48 ^c^	7.47 ± 1.85 ^c^	1.13 ± 0.31 ^a^	33.72 ± 0.81 ^d^	8.23 ± 0.07 ^a^	54.22 ± 0.03 ^d^
W-4	9.95 ± 0.44 ^c^	7.12 ± 0.96 ^c^	1.23 ± 0.08 ^d^	37.76 ± 2.13 ^c^	8.20 ± 0.28 ^b^	56.65 ± 0.10 ^b^
W-5	11.07 ± 0.41 ^a^	6.72 ± 1.54 ^d^	1.17 ± 0.11 ^c^	40.68 ± 0.94 ^f^	8.15 ± 0.07 ^b^	56.64 ± 0.09 ^b^
W-6	11.55 ± 0.67 ^a^	6.12 ± 1.35 ^d^	1.08 ± 0.07 ^d^	41.41 ± 3.55 ^a^	8.16 ± 0.19 ^b^	59.07 ± 0.04 ^a^
W-7	10.59 ± 0.13 ^b^	5.62 ± 0.33 ^d^	1.10 ± 0.17 ^c^	39.58 ± 0.70 ^h^	8.22 ± 0.11 ^a^	54.26 ± 0.08 ^d^
C-1	10.36 ± 0.61 ^b^	5.12 ± 2.59 ^g^	1.06 ± 0.05 ^d^	29.04 ± 2.79 ^b^	8.15 ± 0.07 ^b^	55.04 ± 0.14 ^c^

Notes: Different letters represent a significant difference among multiple groups (*p* < 0.05). *: Extraction yield refers to the total weight of extracted solid from 10 g of tea samples after freeze-drying.

**Table 2 foods-11-03376-t002:** Frequencies of bands of some tea polysaccharides in Fourier-transform infrared spectra, with assigned classification, functional group, and the mode of vibration [46,48,49,50,51,52,53,54].

No.	Frequency (cm^−1^)	Classification	Functional Group	Mode of Vibration	Pile-Fermentation Stages
1	3414–3442	O-H stretching	O−H	Stretching	w-0~c-1
2	2929–2949	Asymmetric C-H stretching	-CH,−CH2	Stretching vibration	w-0~c-1
2	2859–2882	Aldehyde stretching	C−H	Stretching vibration	w-0, w-1, w-3, w-5
3	1609–1638	Carbonyl group	C=O	Stretching	w-0~c-1
4	1419–1436	Asymmetric and antisymmetric C-O stretching	O−C=O	Stretching vibration	w-0~c-1
5	1327	C-O stretching	OCH3	Angle vibration	w-0, w-2, w-4~w-7
6	1239–1250	Pyranose form	C−O−C, C−OH	Stretching vibration	w-0~w-5, w-7, c-1
7	1080–1107	pyranoside	C−O−O	Stretching vibration	w-0~c-1
8	1026–1053	Sugar rings	C−O−O	Stretching vibration	w-0~c-1
9	768–822	C-H bending	−CH, −CH2	Bending vibration	w-0~c-1
10	574–665	α-configuration	C−OH	Presence of α-configuration	w-2~c-1

**Table 3 foods-11-03376-t003:** The main saccharide composition of tea polysaccharides at different pile-fermentation stages of post-fermented tea.

No.	Type	Compound Name	Adduct	Formula	Retention Time	*m*/*z*	Relative Percentage Proportions (%)								
							W-0	W-1	W-2	W-3	W-4	W-5	W-6	W-7	C-1
1	Monosaccharides	d-arabinose	[M-H]- +	C_5_H_10_O_5_	0.644	151.06229	0.05	0.07	0.15	0.23	0.24	0.63	0.75	0.69	0.68
2		l-rhamanose	[M+H]+	C_6_H_12_O_5_	1.814	164.16	0.01	0.06	0.08	0.18	0.19	0.56	0.59	0.66	0.38
3		d-Glucose	[M-H]-	C_6_H_12_O_6_	0.651	179.06084	2.86	3.22	4.3	7.27	8.63	8.84	7.62	6.41	3.18
4		d-Mannose	[M+Na] +	C_6_H_12_O_6_	0.659	180.1564	0.09	0.15	0.38	0.59	0.71	0.83	0.73	0.52	0.39
5		Mannitol	[M+H] +	C_6_H_14_O_6_	0.651	183.0844	0.08	0.09	0.13	0.27	0.39	0.46	0.59	0.87	1.02
6		d-Glucosamine	[M+H]+	C_6_H_13_NO_5_	0.653	180.08598	0.6	0.82	1.61	1.14	1.07	0.94	0.79	0.64	0.21
7		Galactaric acid	[M-H]-	C_6_H_10_O_8_	0.589	209.0363	0.01	0.19	0.1	0.01	0.13	0.39	0.42	0.47	0.41
8		2-Dehydro-d-gluconic acid	[M-H]-	C_12_H_24_O_9_	0.612	193.0385	0.78	2.3	5.32	1.18	2.25	4.70	0.73	0.93	1.46
9		Gluconic acid	[M-H]-	C_6_H_12_O_7_	0.619	195.0581	1.94	4.34	2.65	2.42	6.4	3.62	1.43	1.33	5.50
10		d-Arabitol	[M-H]-	C_5_H_12_O_5_	0.644	151.0623	0.18	0.29	0.3	0.37	0.69	0.74	0.79	0.74	0.87
11		1,5-Anhydro-d-glucitol	[2M+H]+	C_6_H_12_O_5_	0.65	165.0733	0.04	0.15	0.11	0.36	0.25	0.37	0.49	0.30	0.13
12	Oligosaccharides	Sucrose	M-H]-	C_12_H_22_O_11_	0.665	341.1095	9.09	4.78	2.84	1.58	1.58	1.52	1.35	0.98	0.94
13		beta-Maltose	[M+H]+	C_12_H_22_O_11_	0.668	343.1214	1.53	0.99	0.62	0.07	0.24	0.43	0.75	0.69	0.08
14		Raffinose	M-H]-	C_18_H_32_O_16_	0.668	503.1601	3.94	1.96	1.48	0.4	0.08	1.95	1.4	0.77	0.76
15		Maltotriose	[M+Na]+	C_18_H_32_O_16_	0.69	527.1572	2.61	2.06	0.64	0.18	0.18	0.4	0.39	0.26	0.08
16		Stachyose	M-H]-	C_24_H_42_O_21_	0.663	665.2165	0.26	0.24	0.2	0.15	0.11	0.07	0.04	0.03	0.01
17	Monosaccharide and oligosaccharide derivatives	Glucose 6-phosphate	M-H]-	C_6_H_13_O_9_P	0.59	259.0254	0.17	0.59	0.63	0.95	1.29	1.69	1.37	1.17	0.62
18		α-d-Galactose 1-phosphate	M-H]-	C_6_H_13_O_9_P	9.193	259.1914	0.06	0.11	0.13	0.46	0.64	0.82	0.71	0.73	0.21
20		d-Fructofuranose 6-phosphate	[M-H]-	C_6_H_13_O_9_P	4.244	259.0333	0.14	0.39	0.31	0.06	0.27	0.54	0.63	0.84	0.28
21		1,6-Digalloyl-beta-d-glucopyranose	[M-H]-	C_20_H_20_O_14_	3.17	483.0781	0.26	0.27	0.47	0.53	0.63	0.81	0.94	0.71	0.79
22		1,3,6-Tri-*O*-galloyl-beta-d-glucose	[M-H]-	C_27_H_24_O_18_	4.057	635.0873	0.08	0.07	0.1	0.54	0.56	0.66	0.59	0.61	0.59
23		1,2,3-Tri-*O*-galloyl-beta-d-glucose	[M-H]-	C_27_H_24_O_18_	4.837	635.0877	0.02	0.06	0.12	0.13	0.05	0.1	0.04	0	0.7

## Data Availability

Data are available within the article.

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
