# Peer review of "Dynamic Analysis of Physicochemical Properties and Polysaccharide Composition during the Pile-Fermentation of Post-Fermented Tea"

_foods, 2022, doi:10.3390/foods11213376_

Round 1
Reviewer 1 Report
The paper titled “Dynamic analysis of physicochemical properties and polysaccharide compositions during the pile-fermentation of post-fermented tea” represent an interesting and also well-writing study. In my opinion, the paper can be considered for the publication on this Journal. I propose only some advice:
1- Authors must summarize the abstract section (max 200 words), see template
2- I ask to the authors to check the english form.
3- You should apply the same style reported in the template of the Journal, and check the spaces between words.
4- Improve the quality of all the figures.
5- Arrange the table 3 better

Author Response
The paper titled “Dynamic analysis of physicochemical properties and polysaccharide compositions during the pile-fermentation of post-fermented tea” represent an interesting and also well-writing study. In my opinion, the paper can be considered for the publication on this Journal. I propose only some advice:
Question 1: Authors must summarize the abstract section (max 200 words), see template
Answer: Thank you for your suggestions and we have summarized and condensed the summary section.
Question 2: I ask to the authors to check the English form.
Answer: We have checked the English form as requested by the reviewer.
Question 3: You should apply the same style reported in the template of the Journal, and check the spaces between words.
Answer: We have reformatted the changes to the same style as reported in the Journal and checked for Spaces between words.
Question 4: Improve the quality of all the figures.
Answer: We have increased the picture resolution.
Question 5: Arrange the table 3 better
Answer: We have arranged the table 3 by the reviewer.
Reviewer 2 Report
The manuscript submitted by Luo et al. describes the changes in saccharide content changes during fermentation of tea leaves. The plans and experiments sound, and the results are interesting.
Suggestions:
Figures: It may be only for the documents of review purpose, but the resolution of figures are too low.
Table 1: The reviewer assumes the extraction yield is the total weight of extracted solid after freeze-drying. It starts from 50g exactly, and other substances are given mg/g ratio. These presentation is confusing. The reviewer rather suggests to give those numbers with the absolute amounts. For example W-0 sample should be given 865 mg, 118 mg, 3.4 mg, neutral sugar, 70.8 mg, 51.35%. Also another suggestion is to add the calculated amount of polysaccharides. In the case of W-0, it would be 672.8 mg. Another issue in this table is resolution. Does the method really have such fine resolution? I.e., can measure down to 0.001 mg? The reviewer strongly doubts they used an instrument to measure such fine resolution, and also these fine resolution does not mean anything. And lastly the percentage in this table has 0.01% resolution. As the reviewer doubts the resolution of weighing, this much of digits in percentage figures are not required or rather misrepresenting.
Figure 7: I believe Panel a use the same data of blue bars in Panel b. Also this makes some confusions. What are the differences in two panels? What do they like to argue by giving the same data in different formats?
Author Response
Question 1: Figures: It may be only for the documents of review purpose, but the resolution of figures are too low.
Answer: We are very sorry for our careless error and have now increased the picture resolution.
Question 2: Table 1: The reviewer assumes the extraction yield is the total weight of extracted solid after freeze-drying. It starts from 50g exactly, and other substances are given mg/g ratio. This presentation is confusing. The reviewer rather suggests giving those numbers with the absolute amounts. For example, W-0 sample should be given 865 mg, 118 mg, 3.4 mg, neutral sugar, 70.8 mg, 51.35%.
Answer: We are very sorry for the confusion caused by our mistakes in our work. According to your suggestion, we have carefully checked. Now we have updated the content of Table 1 and unified the units.
Question 3: Also, another suggestion is to add the calculated amount of polysaccharides. In the case of W-0, it would be 672.8 mg.
Answer: We have added the calculation of polysaccharides in the revised manuscript.
Question 4: Another issue in this table is resolution. Does the method really have such fine resolution? I.e., can measure down to 0.001 mg? The reviewer strongly doubts they used an instrument to measure such fine resolution, and these fine resolution does not mean anything. And lastly the percentage in this table has 0.01% resolution. As the reviewer doubts the resolution of weighing, this much of digits in percentage figures are not required or rather misrepresenting.
Answer: Regarding another question you raised, we make the following explanation: the dispute may be caused by our wrong presentation of the data in Table 1. The data listed in the table are valid data with two decimal places obtained by calculation, rather than the original data obtained by instrument measurement. Sorry again for our carelessness and hope you can give us new suggestions on the revised manuscript we submitted.
Question 5: Figure 7: I believe Panel a use the same data of blue bars in Panel b. Also, this makes some confusions. What are the differences in two panels? What do they like to argue by giving the same data in different formats?
Answer: We regret that you cannot understand this figure due to the vagueness of our presentation, please allow me to introduce the picture to you: where a show the DPPH radical scavenging capacity of tea polysaccharide at different pile-fermentation stages at a concentration of 1.0mg/ml, we selected several representative tea polysaccharides at different stages and measured their DPPH radical scavenging activity at different concentrations (b). So, a stand of different pile-fermentation stages, b shows different concentrations. The DPPH radical activity of tea polysaccharides during pile-fermentation stages can be clearly known by A and B.
Reviewer 3 Report
Dear author(s):
Dynamic analysis of physicochemical properties and polysaccharide compositions during the pile-fermentation of post-fermented tea
After an exhaustive revision, the manuscript is Reconsider after major revision (control missing in some experiments). In general, the study is closely connected to the journal's objectives. The study is very interesting. The English is good. The introduction is good. The section materials and methods need a Figure to understand the experiments. The section results and discussion is poor, and in turn, it is very confuse, since it is difficult to see the description of the results, the explication of the results, comparison with other studies, and explication (discussion) of the results obtained with respect to other studies is poor. Additionally, the authors need to add lines on comparison with other studies, and explication (discussion) of the results obtained with respect to other studies.
In the following pages, I give a detailed revision of the manuscript.
ABSTRACT
The abstract is good. However, the authors need to add numerical results.
1. INTRODUCTION
The introduction is very clear, concise and precise, with good English, and it has updated references until 2022, and it is good.
What is the meaning of dynamic analysis? The authors need to add some lines.
2. MATERIALS AND METHODS
General comments
This section is clear. The English is good. The authors must add a Figure that represents all the complete methodology. This Figure will help to understand the methodology.
2.6. Analysis of polysaccharide compositions by ultra-high performance liquid chromatography-quadrupole-time of flight tandem mass spectrometry (UHPLC-Q-TOF-MS/MS)
What is the reference?
3. RESULTS AND DISCUSSION
The section of “Results and Discussion” is characterized by a very detailed description of the results, explication of the results, comparison with other studies, and explication (discussion) of the results obtained with respect to other studies.
3.1. Analysis on physicochemical properties of post-fermented tea during the pile-fermentation
Lines 237-239
The authors need to add more information, it is an easy analysis, but the authors need to add explication of the results, comparison with other studies, and explication (discussion) of the results obtained with respect to other studies.
Lines 239-246
The description of the results is good, but the authors need to add information on statistical differences, the explication of the results is good, the authors need to add comparison with other studies, and explication (discussion) of the results obtained with respect to other studies.
Lines 247-261
The description of the results is good, but the authors need to add information on statistical differences.
The explication of the results: “The changes of L*, a*, b* values of tea samples were significantly correlated with the changes of physical and chemical components of tea samples [35-36]. With the content decrease of tea polyphenols, amino acids, catechins and the increase of theabrownins content, the L* value of tea samples gradually decreased, while significant negative correlations were found between a*, b* value and the content of main component (such as tea polyphenols, amino acids, catechins) of tea.” is poor, since it is very general.
The authors need to add comparison with other studies, and explication (discussion) of the results obtained with respect to other studies.
Lines 262-269
The description of the results is poor, but the authors need to add information on statistical differences.
The explication of the results: “which may be closely related to a series of biochemical reactions such as degradation and oxidation under high temperature and humidity or the action of microbial enzymes during the pile-fermentation process of post-fermented tea [37].” is poor, since it is very general.
The authors need to add comparison with other studies, and explication (discussion) of the results obtained with respect to other studies.
Lines 279-302
The lines present description of the results and explication of the results (without reference). However, these points are mixed, and it is very difficult to understand. Therefore, the authors need to order the lines. Additionally, the authors need to add comparison with other studies, and explication (discussion) of the results obtained with respect to other studies.
3.2. Analysis on number of microorganisms during the pile-fermentation of post-fermented tea
The changes of microbial population during the pile-fermentation process were shown in the figure 1.
Or Figure 3?
The description of the results is good, but the authors need to add information on statistical differences.
The explication of the results: “and their importance to the formation of post-fermented tea quality in the process of pile-fermentation has been confirmed by many studies [43].” Is very poor.
The authors need to add comparison with other studies, and explication (discussion) of the results obtained with respect to other studies.
3.3. Chemical components of polysaccharides in post-fermented tea during the pile-fermentation
Lines 326-335
The description of the results is good, but the authors need to add information on statistical differences.
The explication of the results: “which indicated that the formation of polysaccharides in tea may be affected by microbial metabolism and secretion of extracellular enzymes [44] during the pile-fermentation process of tea samples.” is very poor.
Lines 336-348
The description of the results is good, but the authors need to add information on statistical differences.
The authors need to add the explication of the results.
The comparison with other studies “which were consistent with the result of polysaccharide compositions studied by Mao et al. [27]” is good, but the authors need to add more details on the study.
The explication (discussion) of the results obtained with respect to other studies “and indicated that polysaccharide molecules were easy to form more stable macromolecules with tea polyphenol, amino acids through hydrogen bond or other links.” is good.
Lines 356-376
These lines are correct, with description of the results and the explication of the results. However, the authors need to add comparison with other studies and explication (discussion) of the results obtained with respect to other studies, since “with the chemical composition of TPS reported by the previous studies [47-48].” is insufficient.
** What is the subsection 3.4?**
3.5. Mass spectrometry analysis of tea polysaccharide components during the pile-fermentation of post-fermented tea
The description of the results is good.
The explication of results is good, but the authors need to add references.
However, the authors need to add comparison with other studies and explication (discussion) of the results obtained with respect to other studies, since “In the process of tea pile-fermentation, microorganisms may be involved in the conversion of polysaccharides such as cellulose [43] to oligosaccharide, oligosaccharide to monosaccharide, and the derivation (such as phosphorylation) of oligosaccharide and monosaccharide through enzymatic action for their utilization of carbohydrates.” is insufficient.
The authors need to add explication of results, comparison with other studies and explication (discussion) of the results obtained with respect to other studies for the results on Figure 5 and Figure 6.
3.6. DPPH free radical scavenging activity of tea polysaccharides during the pile-fermentation of post-fermented tea
The description of the results is good, but the authors need to add information on statistical differences.
The explication of results is poor, without references.
The authors need to add comparison with other studies and explication (discussion) of the results obtained with respect to other studies.
3.7. Correlation analysis between compositions, biological activities of tea polysaccharide and physicochemical characteristics during the pile-fermentation of post-fermented tea
This subsection is good.
4. CONCLUSIONS
The conclusions should not present results. The conclusions are emphatic points where the advantages, disadvantages, perspectives, and challenges, among others, of the study carried out are indicated.

Author Response
ABSTRACT:
Question 1: The abstract is good. However, the authors need to add numerical results.
Answer: Thank you for your affirmation and suggestion. We have added some numerical results to the abstract and shown them with a yellow background.
INTRODUCTION:
Question 2: The introduction is very clear, concise and precise, with good English, and it has updated references until 2022, and it is good. What is the meaning of dynamic analysis? The authors need to add some lines.
Answer: Thank you for your recognition. In response to your suggestion, we have added a sentence related to the meaning of dynamic analysis in line 110-114 of the revised manuscript.
MATERIALS AND METHODS
Question 3: This section is clear. The English is good. The authors must add a Figure that represents all the complete methodology. This Figure will help to understand the methodology.
Answer: Thank you for pointing this out and we have added a Figure that represents all the complete methodology.
2.6. Analysis of polysaccharide compositions by ultra-high performance liquid chromatography-quadrupole-time of flight tandem mass spectrometry (UHPLC-Q-TOF-MS/MS)
Question 4: What is the reference?
Answer: It has a reference of [30], and we have added a reference [31] for this purpose.
RESULTS AND DISCUSSION
The section of “Results and Discussion” is characterized by a very detailed description of the results, explication of the results, comparison with other studies, and explication (discussion) of the results obtained with respect to other studies.
3.1Analysis on physicochemical properties of post-fermented tea during the pile-fermentation
Question 5: Lines 237-239:The authors need to add more information, it is an easy analysis, but the authors need to add explication of the results, comparison with other studies, and explication (discussion) of the results obtained with respect to other studies.
Answer: As suggested by the reviewer, we have added explication of the results in line 249-252 of the revised manuscript.
Question 6: Lines 239-246: The description of the results is good, but the authors need to add information on statistical differences, the explication of the results is good, the authors need to add comparison with other studies, and explication (discussion) of the results obtained with respect to other studies.
Answer: Thank you for pointing this out and we have added information on statistical differences.
Question 7: Lines 247-261: The description of the results is good, but the authors need to add information on statistical differences.
Answer: Thank you for pointing this out and we have added information on statistical differences.
Question 8: The explication of the results: “The changes of L*, a*, b* values of tea samples were significantly correlated with the changes of physical and chemical components of tea samples [35-36]. With the content decrease of tea polyphenols, amino acids, catechins and the increase of theabrownins content, the L* value of tea samples gradually decreased, while significant negative correlations were found between a*, b* value and the content of main component (such as tea polyphenols, amino acids, catechins) of tea.” is poor, since it is very general. The authors need to add comparison with other studies, and explication (discussion) of the results obtained with respect to other studies.
Answer: As suggested by the reviewer, we have added comparisons with other studies to the revised manuscript.
Question 9: Lines 262-269: The description of the results is poor, but the authors need to add information on statistical differences.
Answer: Thank you for pointing this out and we have added information on statistical differences.
Question 10: The explication of the results: “which may be closely related to a series of biochemical reactions such as degradation and oxidation under high temperature and humidity or the action of microbial enzymes during the pile-fermentation process of post-fermented tea [37].” is poor, since it is very general. So, the authors need to add comparison with other studies, and explication (discussion) of the results obtained with respect to other studies.
Answer: As suggested by the reviewer, we have added comparisons with other studies to the revised manuscript.
Question 11: Lines 279-302: The lines present description of the results and explication of the results (without reference). However, these points are mixed, and it is very difficult to understand. Therefore, the authors need to order the lines. Additionally, the authors need to add comparison with other studies, and explication (discussion) of the results obtained with respect to other studies.
Answer: We sincerely thank the reviewer for careful reading and suggestions, we have sorted the rows as you suggested.
3.2. Analysis on number of microorganisms during the pile-fermentation of post-fermented tea
Question 12: The changes of microbial population during the pile-fermentation process were shown in the figure 1?Or Figure 3?
Answer: We are sorry for our carelessness. Thank you for your warning. Line 314, the statements of “figure 1.” were corrected as “figure 3.”
Question 13: The description of the results is good, but the authors need to add information on statistical differences.
Answer: Thank you for pointing this out and we have added information on statistical differences.
Question 14: The explication of the results: “and their importance to the formation of post-fermented tea quality in the process of pile-fermentation has been confirmed by many studies [43].” Is very poor. The authors need to add comparison with other studies, and explication (discussion) of the results obtained with respect to other studies.
Answer: Thank you for pointing this out. The reviewer is correct, and we have added comparisons with other studies to the revised manuscript.
3.3. Chemical components of polysaccharides in post-fermented tea during the pile-fermentation
Question 15: Lines 326-335: The description of the results is good, but the authors need to add information on statistical differences.
Answer: Thank you for pointing this out and we have added information on statistical differences.
Question 16: The explication of the results: “which indicated that the formation of polysaccharides in tea may be affected by microbial metabolism and secretion of extracellular enzymes [44] during the pile-fermentation process of tea samples.” is very poor.
Answer: In response to your suggestions, we have adjusted the interpretation of the results, as shown in line 358-362 of the revised manuscript with yellow background.
Question 17: Lines 336-348: The description of the results is good, but the authors need to add information on statistical differences.
Answer: Thank you for pointing this out and we have added information on statistical differences.
Question 18: The authors need to add the explication of the results.
Answer: we have added the explication of the results.
Question 19: The comparison with other studies “which were consistent with the result of polysaccharide compositions studied by Mao et al. [27]” is good, but the authors need to add more details on the study.
Answer: Thank you for pointing this out and we have added some information on the study of Mao et al.
Question 20: The explication (discussion) of the results obtained with respect to other studies “and indicated that polysaccharide molecules were easy to form more stable macromolecules with tea polyphenol, amino acids through hydrogen bond or other links.” is good.
Answer: Thank you for your encouragement.
Question 21: Lines 356-376: These lines are correct, with description of the results and the explication of the results. However, the authors need to add comparison with other studies and explication (discussion) of the results obtained with respect to other studies, since “with the chemical composition of TPS reported by the previous studies.
Answer: Thank you very much for your suggestion. By comparing with other studies, we have analyzed the infrared spectrum data and concluded that all samples have similar infrared spectra, and there are characteristic spectra of protein and pyran-glycosides form and sugar rings. In addition, we added that “we added that these results showed that all tea polysaccharides are composed of sugars and proteins, which consistent with Wei et al. [47] proposed that the essence of tea polysaccharide is a glycoprotein, whose protein is closely bound to the sugar chain by N- or O- covalent bond and kang [48] and Chen’s [49] study thatβ-pyranoside-linked, protein-binding polysaccharides.
Question 22: What is the subsection 3.4?
Answer: I'm sorry that our description was confusing to you. The subsection shows Infrared spectroscopic analysis of polysaccharides in post-fermented tea during the pile-fermentation, The FT-IR spectra of polysaccharides extracted from tea samples at different pile-fermentation stage of post-fermented tea were shown in Figure 4, and the as assignment of the main FT-IR bands of polysaccharides was listed in Table 2. The spectral band wave number corresponding to some groups or chemical bonds in different compounds is basically fixed or only changes in a small band range. Therefore, the organic functional groups in tea polysaccharide samples can be determined by infrared spectroscopy, and the components of tea polysaccharide can be determined simply.
3.5. Mass spectrometry analysis of tea polysaccharide components during the pile-fermentation of post-fermented tea
Question 23: The description of the results is good.
Answer: Thank you for your encouragement.
Question 24: The explication of results is good, but the authors need to add references.
Answer: In view of your suggestion, we have added the reference [31] in the section.
Question 25: However, the authors need to add comparison with other studies and explication (discussion) of the results obtained with respect to other studies, since “In the process of tea pile-fermentation, microorganisms may be involved in the conversion of polysaccharides such as cellulose [43] to oligosaccharide, oligosaccharide to monosaccharide, and the derivation (such as phosphorylation) of oligosaccharide and monosaccharide through enzymatic action for their utilization of carbohydrates.” is insufficient.
Answer: According to your suggestions, we have supplemented relevant content in the revised manuscript.
Question 26: The authors need to add explication of results, comparison with other studies and explication (discussion) of the results obtained with respect to other studies for the results on Figure 5 and Figure 6.
Answer: Thank you for point this out, we have some case need and your specification, the fermented tea after fermentation of tea polysaccharide component in related studies haven't seen yet, so we can only try to get this research result to make detailed description (by PCA and heat map analysis, we find the polysaccharide of the similarity between the samples, The whole W pile process can be roughly divided into fermentation early, middle and late three times, based on the simple sugars, chitosan and derivatives of three components, different in every W pile of phase change, monosaccharide content gradually increased, in a pile of Woodward oligonucleotides glycan slowly decrease, composition transformation mainly in ~ W W - 1-7 times), of course, We also hope that our study can complement the gap in this area and provide a certain reference for subsequent research
3.6. DPPH free radical scavenging activity of tea polysaccharides during the pile-fermentation of post-fermented tea
Question 27: The description of the results is good, but the authors need to add information on statistical differences.
Answer: Thank you for pointing this out and we have added information on statistical differences.
Question 28: The explication of results is poor, without references, And the authors need to add comparison with other studies and explication (discussion) of the results obtained with respect to other studies
Answer: We have added references and comparisons with others, as shown in the yellow background of section 3.6 of the revised manuscript.
3.7. Correlation analysis between compositions, biological activities of tea polysaccharide and physicochemical characteristics during the pile-fermentation of post-fermented tea
Question 29: This subsection is good.
Answer: I am very glad to have your recognition of our work
- CONCLUSIONS
Question 30: The conclusions should not present results. The conclusions are emphatic points where the advantages, disadvantages, perspectives, and challenges, among others, of the study carried out are indicated.
Answer: According to your comments, we have adjusted the conclusion, please see the manuscript for details.
Reviewer 4 Report
The manuscript “Dynamic analysis of physicochemical properties and polysaccharide compositions during the pile-fermentation of post-fermented tea” is an investigation into how physicochemical properties and saccharide profile of dark tea during pile-fermentation. The results showed that physical properties and chemical composition changed because of microbial activity during the fermentation. Through the review on this manuscript, I recommend reconsidering the following points:
- Section 2.8: please provide more information about ANOVA and chemometric tools (PCA and heatmap). SD was standard deviation?
- Please improve the resolution of all the figures.
- I feel uncertainty about the polysaccharide results obtained from QTOF MS. The report on high resolution MS/MS without information about adducts, fragmentation ions, mass error and only two decimals for m/z are insufficient and inaccurate.
Author Response
Question 1: Section 2.8: please provide more information about ANOVA and chemometric tools (PCA and heatmap). SD was standard deviation?
Answer: We appreciate it very much for this good suggestion, and we have done it according to your ideas. SD is the standard deviation.
Question 2: Please improve the resolution of all the figures
Answer: We are very sorry for our careless mistake, which has now been corrected.
Question 3: I feel uncertainty about the polysaccharide results obtained from QTOF MS. The report on high resolution MS/MS without information about adducts, fragmentation ions mass error and only two decimals for m/z are insufficient and inaccurate.
Answer: Thank you for pointing this out. The high-resolution MS/MS report contains adduct, m/z with only two decimals is the result after we rounded the original data to make the data uniform. The information about adducts has been added to Table 3, which also shows the original data of m/z.
Round 2
Reviewer 3 Report
Dear Author(s)
After an exhaustive revision, the manuscript is Accept in present form. The resubmitted manuscript has been completely improved compared to its previous version. Therefore, the manuscript can be published in “Foods”.
Best regards

Author Response
Question 1: After an exhaustive revision, the manuscript is Accept in present form. The resubmitted manuscript has been completely improved compared to its previous version. Therefore, the manuscript can be published in “Foods”.
Answer: Thank you for your recognition and support.
Reviewer 4 Report
I strongly suggest the authors reconsider m/z of the compounds 1, 2, 4, 5 in Table 3. m/z at 397.345 can't be assigned as mannitol. Those reported values are incorrect. And one more time, at least 4 decimals should be used to present the m/z.
Author Response
Question 1: I strongly suggest the authors reconsider m/z of the compounds 1, 2, 4, 5 in Table 3. m/z at 397.345 can't be assigned as mannitol. Those reported values are incorrect.
Answer: I am sorry for the carelessness in our work. Based on your suggestion, we re-checked the data in Table 3 and corrected the m/z of 1, 2, 4, and 5. The m/z of mannitol was 183.0844, which was also corrected.
Question 2: And one more time, at least 4 decimals should be used to present the m/z.
Answer: Thank you very much for pointing that out. We have modified the m/z column of Table 3. Now they're all at least 4 decimal places.